# REINFORCEMENT LEARNING WITH CHROMATIC NETWORKS

## ABSTRACT

We present a neural architecture search algorithm to construct compact reinforcement learning (RL) policies, by combining ENAS (Vinyals et al., 2015; Pham et al., 2018; Zoph & Le, 2017) and ES (Salimans et al., 2017) in a highly scalable and intuitive way. By defining the combinatorial search space of NAS to be the set of different edge-partitionings (colorings) into same-weight classes, we represent compact architectures via efficient learned edge-partitionings. For several RL tasks, we manage to learn colorings translating to effective policies parameterized by as few as 17 weight parameters, providing $> 90\%$ compression over vanilla policies and 6x compression over state-of-the-art compact policies based on Toeplitz matrices (Choromanski et al., 2018), while still maintaining good reward. We believe that our work is one of the first attempts to propose a rigorous approach to training structured neural network architectures for RL problems that are of interest especially in mobile robotics (Gage, 2002) with limited storage and computational resources.

## 1 INTRODUCTION

Consider a fixed Markov Decision Process (MDP) $\mathcal{M}$ and an agent aiming to maximize its total expected/discounted reward obtained in the environment $\mathcal{E}$ governed by $\mathcal{M}$. An agent is looking for a sequence of actions $a_0, ..., a_{T-1}$ leading to a series of steps maximizing this reward. One of the approaches is to construct a policy $\pi_\theta : \mathcal{S} \to \mathcal{A}$, parameterized by vector $\theta$, which is a mapping from states to actions. Policy $\pi_\theta$ determines actions chosen in states visited by an agent. Such a reinforcement learning (RL) policy is usually encoded as a neural network, in which scenario parameters $\theta$ correspond to weights and biases of a neural network. Reinforcement learning policies $\pi_\theta$ often consist of thousands or millions of parameters (e.g. when they involve vision as part of the state vector) and therefore training them becomes a challenging high-dimensional optimization problem. Deploying such high-dimensional policies on hardware raises additional concerns in resource constrained settings (e.g. limited storage), emerging in particular in mobile robotics (Gage, 2002). The main question we tackle in this paper is the following:

*Are high dimensional architectures necessary for encoding efficient policies and if not, how compact can they be in in practice?*

We show that finding such compact representations is a nontrivial optimization problem despite recent observations that some hardcoded structured families (Choromanski et al., 2018) provide certain levels of compactification and good accuracy at the same time.

We model the problem of finding compact presentations by using a joint objective between the combinatorial nature of the network's parameter sharing profile and the reward maximization of RL optimization. We leverage recent advances in the ENAS (Efficient Neural Architecture Search) literature and theory of pointer networks (Vinyals et al., 2015; Pham et al., 2018; Zoph & Le, 2017) to optimize over the combinatorial component of this objective and state of the art evolution strategies (ES) methods (Choromanski et al., 2018; Salimans et al., 2017; Mania et al., 2018a) to optimize over the RL objective. We propose to define the combinatorial search space to be the the set of different edge-partitioning (colorings) into same-weight classes and construct policies with learned weight-sharing mechanisms. We call networks encoding our policies: *chromatic networks*.

We are inspired by two recent papers: (Choromanski et al., 2018) and (Gaier & Ha, 2019). In the former one, policies based on Toeplitz matrices were shown to match their unstructured counterparts accuracy-wise, while leading to the substantial reduction of the number of parameters from

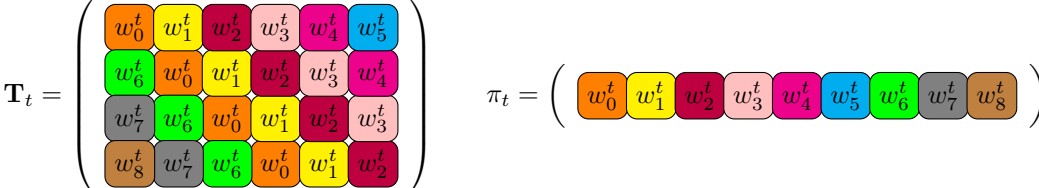

Figure 1: On the left: matrix encoding linear Toeplitz policy at time $t$ for the RL task with 6-dimensional state vector and 4-dimensional action vector. On the right: that policy in the vectorized form. As we see, a policy defined by a matrix with 24 entries is effectively encoded by a 9-dimensional vector.

thousands (Salimans et al., 2017) to hundreds (Choromanski et al., 2018). Instead of quadratic (in sizes of hidden layers), those policies use only linear number of parameters. The Toeplitz structure can be thought of as a parameter sharing mechanism, where edge weights along each diagonal of the matrix are the same (see: Fig. 1). However, this is a rigid pattern that is not learned. We show in this paper that weight sharing patterns can be effectively learned, which further reduces the number of distinct parameters. For instance, using architectures of the same sizes as those in (Choromanski et al., 2018), we can train effective policies for OpenAI Gym tasks with as few as 17 distinct weights. The latter paper proposes an extremal approach, where weights are chosen randomly instead of being learned, but the topologies of connections are trained and thus are ultimately strongly biased towards RL tasks under consideration. It was shown in (Gaier & Ha, 2019) that such weight agnostic neural networks (WANNs) can encode effective policies for several nontrivial RL problems. WANNs replace conceptually simple feedforward networks with general graph topologies using NEAT algorithm (Stanley & Miikkulainen, 2002) providing topological operators to build the network.

Our approach is a middle ground, where the topology is still a feedforward neural network, but the weights are partitioned into groups that are being learned in a combinatorial fashion using reinforcement learning. While (Chen et al., 2015) shares weights *randomly* via hashing, we learn a good partitioning mechanisms for weight sharing.

Our key observation is that *ENAS and ES can naturally be combined in a highly scalable but conceptually simple way*. To give context, vanilla NAS (Zoph & Le, 2017) for classical supervised learning setting (SL) requires a large population of 450 GPU-workers (child models) all training one-by-one, which results in many GPU-hours of training. ENAS (Pham et al., 2018) uses weight sharing across multiple workers to reduce the time, although it can reduce computational resources at the cost of the variance of the controller's gradient. Our method solves both issues (fast training time and low controller gradient variance) by leveraging a large population of much-cheaper CPU workers (300) increasing the effective batch-size of the controller, while also training the workers simultaneously via ES. This setup is not possible in SL, as single CPUs cannot train large image-based classifiers in practice. Furthermore, this magnitude of scaling by numerous workers can be difficult with policy gradient or Q-learning methods as they can be limited by GPU overhead due to exact-gradient computation.

We believe that our work is one of the first attempts to propose a flexible, rigorous approach to training compact neural network architectures for RL problems. Those may be of particular importance in mobile robotics (Gage, 2002) where computational and storage resources are very limited. We also believe that this paper opens several new research directions regarding structured policies for robotics.

## 1.1 BACKGROUND AND RELATED WORK

**Network Architecture Search:** The subject of this paper can be put in the larger context of Neural Architecture Search (NAS) algorithms which recently became a prolific area of research with already voluminous literature (see: (Elsken et al., 2019) for an excellent survey). Interest in NAS algorithms started to grow rapidly when it was shown that they can design state-of-the-art architectures for image recognition and language modeling (Zoph & Le, 2017). More recently it was shown that NAS

network generators can be improved to sample more complicated connectivity patterns based on random graph theory models such as Erdos-Renyi, Barabasi-Albert or Watts-Strogatz to outperform human-designed networks, e.g. ResNet and ShuffleNet on image recognition tasks (Xie et al., 2019). However to the best of our knowledge, applying NAS to construct compact RL policy architectures has not been explored before. On a broader scope, there has been little work in applying NAS to general RL policies, partially because unlike SL, RL policies are quite small and thus do not require a search space involving *hyperparameter* primitives such as number of hidden layers, convolution sizes, etc. However, we show that the search space for RL policies can still be quite combinatorial, by examining partitionings of weights in our work.

**Parameterizing Compact Architectures:**   Before NAS can be applied, a particular parameterization of a compact architecture defining combinatorial search space needs to be chosen. That leads to the vast literature on compact encodings of NN architectures. Some of the most popular and efficient techniques regard network sparsification. The sparsity can be often achieved by *pruning* already trained networks. These methods have been around since the 1980s, dating back to Rumelhart (Rumelhart, 1987; Chauvin, 1989; Mozer & Smolensky, 1989), followed shortly by Optimal Brain Damage (Cun et al., 1990), which used second order gradient information to remove connections. Since then, a variety of schemes have been proposed, with regularization (Louizos et al., 2018) and magnitude-based weight pruning methods (Han et al., 2015; See et al., 2016; Narang et al., 2017) increasingly popular. The impact of *dropout* (Srivastava et al., 2014) has added an additional perspective, with new works focusing on attempts to learn sparse networks (Gomez et al., 2019). Another recent work introduced the Lottery Ticket Hypothesis (Frankle & Carbin, 2019), which captivated the community by showing that there exist equivalently sparse networks which can be trained from scratch to achieve competitive results. Interestingly, these works consistently report similar levels of compression, often managing to match the performance of original networks with up to 90% fewer parameters. Those methods are however designed for constructing classification networks rather than those encoding RL policies.

**Quantization:**   Weight sharing mechanism can be viewed from the quantization point of view, where pre-trained weights are quantized, thus effectively partitioned. Examples include (Han et al., 2016), who achieve 49x compression for networks applied for vision using both pruning and weight sharing (by quantization) followed by Huffman coding. However such partitions are not learned which is a main topic of this paper. Even more importantly, in RL applications, where policies are often very sensitive to parameters' weights (Iscen et al., 2018), centroid-based quantization is too crude to preserve accuracy.

**Compact RL Policies**   In recent times there has been increased interest in simplifying RL policies. In particular, (Mania et al., 2018b) demonstrated that linear policies are often sufficient for the benchmark MuJoCo locomotion tasks, while (Cuccu et al., 2019) found smaller policies could work for vision-based tasks by separating feature extraction and control. Finally, recent work found that small, sparse sub-networks can perform better than larger over-parameterized ones (Frankle & Carbin, 2019), inspiring applications in RL (Yu et al., 2019).

Our main contributions are:

1. We propose a highly scalable algorithm by combining ENAS and ES for learning compact representations that learns effective policies with over 92% reduction of the number of neural network parameters (Section 3).
2. To demonstrate the impact (and limitations) of pruning neural networks for RL, we adapt recent algorithms training both masks defining combinatorial structure as well as weights of a deep neural network concurrently (see: (Lenc et al., 2019)). Those achieve state-of-the-art results on various supervised feedforward and recurrent models. We confirm these findings in the RL setting by showing that good rewards can be obtained up to a high level of pruning. However, at the 80-90% level we see a significant decrease in performance which does not occur for the proposed by us chromatic networks (Section 4.1, Section 4.2).
3. We demonstrate that finding efficient weight-partitioning mechanisms is a challenging problem and NAS helps to construct distributions producing good partitionings for more difficult RL environments (Section 4.3).

## 2 THE ARCHITECTURE OF NEURAL ARCHITECTURE SEARCH

The foundation of our algorithm for learning structured compact policies is the class of ENAS methods (Pham et al., 2018). To present our algorithm, we thus need to first describe this class. ENAS algorithms are designed to construct neural network architectures thus they aim to solve combinatorial-flavored optimization problems with exponential-size domains. The problems are cast as MDPs, where a controller encoded by the LSTM-based policy $\pi_{\text{cont}}(\theta)$, typically parameterized by few hundred hidden units, is trained to propose good-quality architectures, or to be more precise: good-quality distributions $\mathcal{D}(\theta)$ over architectures. In standard applications the score of the particular distribution $\mathcal{D}(\theta)$ is quantified by the average performance obtained by trained models leveraging architectures $\mathcal{A} \sim \mathcal{D}(\theta)$ on the fixed-size validation set. That score determines the reward the controller obtains by proposing $\mathcal{D}(\theta)$. LSTM-based controller constructs architectures using softmax classifiers via autoregressive strategy, where controller's decision in step $t$ is given to it as an input embedding at time $t + 1$. The initial embedding that the controller starts with is an empty one.

**Weight Sharing Mechanism:** ENAS introduces a powerful idea of a weight-sharing mechanism. The core concept is that different architectures can be embedded into combinatorial space, where they correspond to different subgraphs of the given acyclic directed base graph **G** (DAG). Weights of the edges of **G** represent the shared-pool $\mathcal{W}_{\text{shared}}$ from which different architectures will inherit differently by activating weights of the corresponding induced directed subgraph. At first glance such a mechanism might be conceptually problematic since, a weight of the particular edge $e$ belonging to different architectures $\mathcal{A}_{i_1}, ... \mathcal{A}_{i_k}$ (see: next paragraph for details regarding weight training) will be updated based on evaluations of all of them and different $\mathcal{A}_i s$ can utilize $e$ in different ways. However it turns out that this is desirable in practice. As authors of (Pham et al., 2018) explain, the approach is motivated by recent work on transfer and multitask learning that provides theoretical grounds for transferring weights across models. Another way to justify the mechanism is to observe that ENAS tries in fact to optimize a distribution over architectures rather than a particular architecture and the corresponding shared-pool of weights $\mathcal{W}_{\text{shared}}$ should be thought of as corresponding to that distribution rather than its particular realizations. Therefore the weights of that pool should be updated based on signals from all different realizations.

**Alternating Optimization:** For a fixed parameterization $\theta$ defining policy $\pi(\theta)$ of the controller (and thus also proposed distribution over architectures $\mathcal{D}(\theta)$), the algorithm optimizes the weights of the models using $M$ architectures: $\mathcal{A}_1, ..., \mathcal{A}_M$ sampled from $\mathcal{D}(\theta)$, where the sampling is conducted by the controller's policy $\pi_{\text{cont}}(\theta)$. Models corresponding to $\mathcal{A}_1, ..., \mathcal{A}_M$ are called *child models*. At iteration $k$ of the weight optimization process, a worker assigned to the architecture $\mathcal{A}_i$ computes the gradient of the loss function $\mathcal{L}_{\mathcal{A}_i, \mathcal{B}_k}$ corresponding to the particular batch $\mathcal{B}_k$ of the training data with respect to the weights of the inherited edges from the base DAG. Since the loss function $\mathcal{L}_{\mathcal{B}_k}$ on $\mathcal{W}_{\text{shared}}$ for a fixed distribution $\mathcal{D}(\theta)$ and batch $\mathcal{B}_k$ at iteration $k$ is defined as an expected loss $\mathbb{E}_{\mathcal{A} \sim \pi(\theta)}[\mathcal{L}_{\mathcal{A}, \mathcal{B}_k}(\mathcal{W}_{\text{shared}}^{\mathcal{A}})]$, where $\mathcal{W}_{\text{shared}}^{\mathcal{A}}$ stands for the projection of the set of shared-pool weights $\mathcal{W}_{\text{shared}}$ into edges defined by the induced subgraph determined by $\mathcal{A}$, its gradient with respect to $\mathcal{W}_{\text{shared}}$ can be estimated via Monte Carlo procedure as:

$$\nabla_{\mathcal{W}_{\text{shared}}} \mathcal{L}_{\mathcal{B}_k}(\mathcal{W}_{\text{shared}}) \sim \frac{1}{M} \sum_{i=1}^{M} \nabla_{\mathcal{W}_{\text{shared}}^{\mathcal{A}_i}} \mathcal{L}_{\mathcal{A}_i, \mathcal{B}_k}(\mathcal{W}_{\text{shared}}^{\mathcal{A}_i}) \tag{1}$$

After weight optimization is completed, ENAS updates the parameters $\theta$ of the controller responsible for shaping the distribution used to sample architectures $\mathcal{A}$. As mentioned before, this is done using reinforcement learning approach, where parameters $\theta$ are optimized to maximize the expected reward $\mathbb{E}_{\mathcal{A} \sim \pi(\theta)}[\mathcal{R}_{\mathcal{W}_{\text{shared}}^{\mathcal{A}}}(\mathcal{A})]$, where this time set $\mathcal{W}_{\text{shared}}$ is frozen and $\mathcal{R}_{\mathcal{W}_{\text{shared}}^{\mathcal{A}}}(\mathcal{A})$ is given as the accuracy obtained by the model using architecture $\mathcal{A}$ and weights from $\mathcal{W}_{\text{shared}}^{\mathcal{A}}$ on the validation set. Parameters $\theta$ are updated with the use of the REINFORCE algorithm (Williams, 1992).

**Controller's Architecture:** The controller LSTM-based architecture uses pointer networks (Vinyals et al., 2015) that utilize LSTM-encoder/decoder approach, where the decoder can look back and forth over input. Thus these models are examples of architectures enriched with attention.

## 3   Towards Chromatic Networks - ENAS for Graph Partitionings

**Preliminaries:**   Chromatic networks are feedforward NN architectures, where weights are shared across multiple edges and sharing mechanism is learned via a modified ENAS algorithm that we present below. Edges sharing a particular weight form the so-called *chromatic class*. An example of the learned partitioning is presented on Fig. 2. Learned weight-sharing mechanisms are more complicated than hardcoded ones from Fig. 1, but lead to smaller-size partitionings. For instance, Toeplitz sharing mechanism for architecture from Subfigure (b) of Fig. 2 requires 103 weight-parameters, while ours: only 17.

**Controller Architecture & Training:**   We use a standard ENAS reinforcement learning controller similar to (Pham et al., 2018), applying pointer networks. Our search space is the set of all possible mappings $\Phi : E \rightarrow \{0, 1, ..., M - 1\}$, where $E$ stands for the set of all edges of the graph encoding an architecture and $M$ is the number of partitions, which the *user* sets. Thus as opposed to standard ENAS approach, where the search space consists of different subgraphs, we instead deal with different colorings/partitionings of edges of a given base graph. The controller learns distributions $\mathcal{D}(\theta)$ over these partitionings. The shared pool of weights $\mathcal{W}_{\mathrm{shared}}$ is a latent vector in $\mathbb{R}^M$, where different entries correspond to weight values for different partitions enumerated from $0$ to $M - 1$. As for standard ENAS, the controller $\pi(\theta)$ consists of a encoder RNN and decoder RNN, in which the encoder RNN is looped over the embedded input data, while the decoder is looped to repeatedly output smaller primitive components of the final output.

In our setting, we do not possess natural numerical data corresponding to an embedding of the inputs which are partition numbers and edges. Therefore we also train an embedding (as part of controller's parameter vector $\theta$) of both using tables: $V_{\mathrm{edge}} : e \rightarrow \mathbb{R}^d$ and $V_{\mathrm{partition}} : \{0, 1, ..., M - 1\} \rightarrow \mathbb{R}^d$.

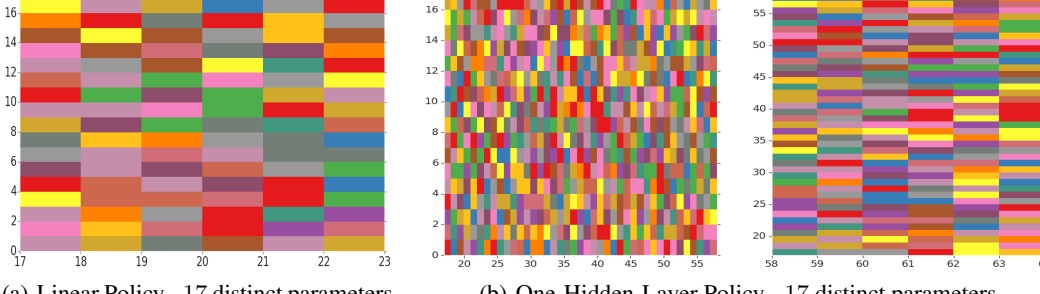

(a) Linear Policy - 17 distinct parameters          (b) One-Hidden-Layer Policy - 17 distinct parameters

Figure 2: On the left: partitioning of edges into distinct weight classes obtained for the linear policy for HalfCheetah environment from OpenAI Gym. On the right: the same, but for a policy with one hidden layer encoded by two matrices. State and action dimensionalities are: $s = 17$ and $a = 6$ respectively and hidden layer for the architecture from (b) is of size $41$. Thus the size of the matrices are: $17 \times 6$ for the linear policy from (a) and: $17 \times 41$, $41 \times 6$ for the nonlinear one from (b).

We denote by $P$ a partitioning of edges and define the reward obtained by a controller for a fixed distribution $\mathcal{D}(\theta)$ produced by its policy $\pi(\theta)$ as follows:

$$\mathcal{R}(\theta) = \mathbb{E}_{P \sim \pi(\theta)}[\mathcal{R}^{\max}_{\mathcal{W}_{\mathrm{shared}}}(P)], \tag{2}$$

where $\mathcal{R}^{\max}_{\mathcal{W}_{\mathrm{shared}}}(P)$ stands for the maximal reward obtained during weight-optimization phase of the policy with partitioning $P$ and with initial vector of distinct weights $\mathcal{W}_{\mathrm{shared}}$. As for the ENAS setup, in practice this reward is estimated by averaging over $M$ workers evaluating independently different realizations $P$ of $\mathcal{D}(\theta)$. The updates of $\theta$ are conducted with the use of REINFORCE.

### 3.1   Weight Updates via ES

As opposed to standard ENAS, where weights for a fixed distribution $\mathcal{D}(\theta)$ generating architectures were trained by backpropagation, we propose to apply recently introduced ES blackbox optimization techniques for RL (Choromanski et al., 2018). For a fixed partitioning $P$, we define the loss $\mathcal{L}_P(\mathcal{W}_{\mathrm{shared}})$ with respect to weights as the negated reward obtained by an agent applying partitioning $P$ and with vector of distinct weights $\mathcal{W}_{\mathrm{shared}}$. That expression as a function of $\mathcal{W}_{\mathrm{shared}}$

is not necessarily differentiable, since it involves calls to the simulator. This also implies that explicit backpropagation is not possible. We propose to instead estimate the gradient of its Gaussian smoothing $\mathcal{L}_P^\sigma(\mathcal{W}_{\text{shared}})$ defined as: $\mathcal{L}_P^\sigma(\mathcal{W}_{\text{shared}}) = \mathbb{E}_{\mathbf{g}\in\mathcal{N}(0,\mathbf{I}_M)}[\mathcal{L}_P(\mathcal{W}_{\text{shared}} + \sigma\mathbf{g})]$ for a fixed smoothing parameter $\sigma >$. We approximate its gradient given by: $\nabla_{\mathcal{W}_{\text{shared}}}\mathcal{L}_P^\sigma(\mathcal{W}_{\text{shared}}) = \frac{1}{\sigma}\mathbb{E}_{\mathbf{g}\in\mathcal{N}(0,\mathbf{I}_M)}[\mathcal{L}_P(\mathcal{W}_{\text{shared}} + \sigma\mathbf{g})\mathbf{g}]$ with the following *forward finite difference* unbiased estimator introduced in (Choromanski et al., 2018):

$$\widehat{\nabla}_{\mathcal{W}_{\text{shared}}}\mathcal{L}_P^\sigma(\mathcal{W}_{\text{shared}}) = \frac{1}{t}\sum_{i=1}^{t}\mathbf{g}_t\left[\frac{\mathcal{L}_P(\mathcal{W}_{\text{shared}} + \sigma\mathbf{g}_t) - \mathbb{E}_{P\sim\pi(\theta)}\left[\mathcal{L}_P(\mathcal{W}_{\text{shared}})\right]}{\sigma}\right] \quad (3)$$

where $\mathbf{g}_1, ..., \mathbf{g}_t$ are sampled independently at random from $\mathcal{N}(0,\mathbf{I}_M)$ and the pivot point is defined as an average loss for a given set of weights $\mathcal{W}_{\text{shared}}$ over partitionings sampled from $\pi(\theta)$.

We note that a subtle key difference here from vanilla ES is by using the pivot point $\mathbb{E}_{P\sim\pi(\theta)}\left[\mathcal{L}_P(\mathcal{W}_{\text{shared}})\right]$ as the expectation over a distribution of partitions $P \sim \pi(\theta)$ rather than a static single-query objective $\mathcal{L}(\mathcal{W})$ used for e.g. a basic Mujoco task. From a broader perspective, this comes from the fact that ES can optimize any objective of the form $\mathbb{E}_{P\sim\mathcal{P}}\left[f(W,P)\right]$ where $P$ is any (possibly discrete) object with distribution $\mathcal{P}$ and $W$ is a continuous weight vector, by assigning both a perturbed weight $W + \sigma\mathbf{g}$ *and* a sampled $P$ to a CPU worker for function evaluation on $f(W + \sigma\mathbf{g}, P)$.

## 4 EXPERIMENTAL RESULTS

The experimental section is organized as follows:

- In Subsection 4.1 we show the limitations of the sparse network approach for compactifying RL policies on the example of state-of-the-art class of algorithms from (Lenc et al., 2019) that aim to simultaneously train weights and connections of neural network architectures. This approach is conceptually the most similar to ours.
- In Subsection 4.2 we present exhaustive results on training our chromatic networks with ENAS on OpenAI Gym and quadruped locomotion tasks. We compare sizes and rewards obtained by our policies with those using masking procedure from (Lenc et al., 2019), applying low displacement rank matrices for compactification as well as unstructured baselines.
- In Subsection 4.3 we analyze in detail the impact of ENAS steps responsible for learning partitions, in particular compare it with the performance of random partitionings.

We provide more experimental results in the Appendix.

### 4.1 LEARNED SPARSE NETWORKS VIA SIMULTANEOUS WEIGHT-TOPOLOGY TRAINING

The mask $m$, drawn from a multinomial distribution, is trained in (Lenc et al., 2019) using ES and element-wise multiplied by the weights before a forward pass. In (Lenc et al., 2019), the sparsity of the mask is fixed, however, to show the effect of pruning, we instead initialize the sparsity at $50\%$ and increasingly reward smaller networks (measured by the size of the mask $|m|$) during optimization. Using this approach on several Open AI Gym tasks, we demonstrate that masking mechanism is capable of producing compact effective policies up to a high level of pruning. At the same time, we show significant decrease of performance at the $80$-$90\%$ compression level, quantifying accurately its limits for RL tasks (see: Fig. 3).

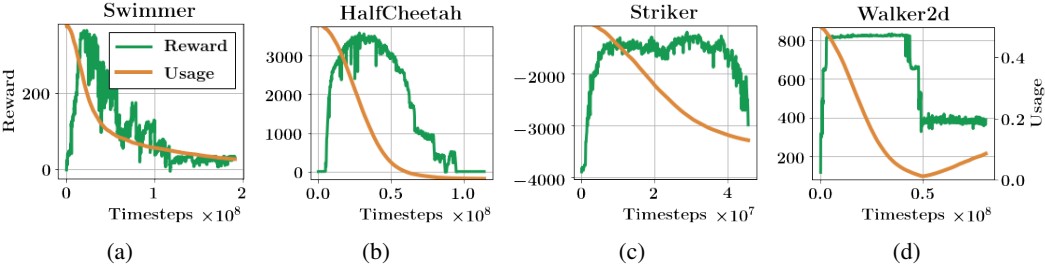

Figure 3: The results from training both a mask $m$ and weights $\theta$ of a neural network with two hidden layers, 41 units each. 'Usage' stands for number of edges used after filtering defined by the mask. At the beginning, the mask is initialized such that $|m|$ is equal to $50\%$ of the total number of parameters in the network.

## 4.2 RESULTS ON CHROMATIC NETWORKS

We perform experiments on the following OpenAI Gym tasks: Swimmer, Reacher, Hopper, HalfCheetah, Walker2d, Pusher, Striker, Thrower and Ant as well as quadruped locomotion task of forward walking from (Iscen et al., 2018). The performance of our algorithm constructing chromatic networks is summarized in Table 1. We tested three classes of feedforward architectures: linear from (Mania et al., 2018a), and nonlinear with one or two hidden layers and $\tanh$ nonlinearities.

We see a general trend that increasing hidden layers while keeping number of partitions fixed, improves performance as well as increasing the number of partitions while keeping the architecture fixed. The relation between the expressiveness of a linear, high-partition policy vs a hidden-layer, low-partition policy is however not well understood. As shown in Table 1, this depends on the environment's complexity as well. For HalfCheetah, the linear 50-partition policy performs better than a hidden layer 17-partition policy, while this is reversed for for the Minitaur.

In Table 2 we directly compare chromatic networks with a masking approach as discussed in Section 4.1, as well as other structured policies (Toeplitz from (Choromanski et al., 2018) and circulant) and the unstructured baseline. In all cases we use the same hyper-parameters, and train until convergence for five random seeds. For masking, we report the best achieved reward with $> 90\%$ of the network pruned, making the final policy comparable in size to the chromatic network. For each class of policies, we compare the number of weight parameters used ("# of weight-params" field), since the compactification mechanism does not operate on bias vectors. We also record compression in respect to unstructured networks in terms of the total number of parameters ("# compression" field). This number determines the reduction of sampling complexity with respect to unstructured networks (which is a bottleneck of ES training), since the number of RL blackbox function F queries needed to train/up-train the policy is proportional to the total number of weights and biases of the corresponding network.

Finally, for a working policy we report total number of bits required to encode it assuming that real values are stored in the $\mathrm{float}$ format. Note that for chromatic and masking networks this includes bits required to encode a dictionary representing the partitioning. Further details are given in the Appendix (Section E). Top two performing networks for each environment are in bold. Chromatic networks are the only to provide big compression and quality at the same time across all tasks.

**Inference Time:** Similarly to Toeplitz, chromatic networks also provide computational gains. Using improved version of the mailman-algorithm (Liberty & Zucker, 2009), matrix-vector multiplication part of the inference can be run on the chromatic network using constant number of distinct weights and deployed on real hardware in time $O(\frac{mn}{\log(\max(m,n))})$, where $(m, n)$ is the shape of the matrix.

## 4.3 FURTHER ANALYSIS: RANDOM PARTITIONINGS VERSUS ENAS
A natural question to ask is whether ENAS machinery is required or maybe random partitioning is good enough to produce efficient policies. To answer it, we trained joint weights for fixed population of random partitionings without NAS, as well as with random NAS controller.

| Environment | Dimensions | Architecture | Partitions | Mean Reward | Max Reward |
|---|---|---|---|---|---|
| Swimmer | (8,2) | L | 8 | 97 | 365 |
| Reacher | (11,2) | L | 11 | -144 | -6 |
| Hopper | (11,3) | L | 11 | 216 | 999 |
| Hopper | (11,3) | H41 | 11 | 247 | 3408 |
| HalfCheetah | (17,6) | L | 17 | 1812 | 3653 |
| HalfCheetah | (17,6) | L | 50 | 1383 | 4318 |
| HalfCheetah | (17,6) | H41 | 17 | 2148 | 3779 |
| HalfCheetah | (17,6) | H41, H41 | 17 | 3036 | 5285 |
| Walker2d | (17,6) | H41 | 17 | 1943 | 3695 |
| Pusher | (23,7) | H41 | 23 | -419 | -144 |
| Striker | (23,7) | H41 | 23 | -1926 | -248 |
| Thrower | (23,7) | H41 | 23 | -1651 | -61 |
| Ant | (111,8) | H41, H41 | 50 | 1047 | 1440 |
| Minitaur | (7, 13) | L | 13 | 4.84 | 7.2 |
| Minitaur | (7, 13) | L | 50 | 6.08 | 7.91 |
| Minitaur | (7, 13) | H41 | 13 | 7.12 | 9.34 |

Table 1: Statistics for training chromatic networks. For mean reward, we take the average over 301 worker rollout-rewards for each step, and output the highest average over all timesteps. For max reward, we report the maximum reward ever obtained during the training process by any worker. "L", "H41" and "H41, H41" stand for: linear policy, policy with one hidden layer of size 41 and policy with two such hidden layers respectively.

| Environment | Architecture | Reward | # weight-params | compression | # bits |
|---|---|---|---|---|---|
| Striker | Chromatic | -248 | 23 | 95% | 8198 |
| | Masked | -967 | 25 | 95% | 8262 |
| | Toeplitz | -129 | 110 | 88% | 4832 |
| | Circulant | **-120** | 82 | 90% | 3936 |
| | Unstructured | **-117** | 1230 | 0% | 40672 |
| HalfCheetah | Chromatic | **3779** | 17 | 94% | 6571 |
| | Masked | **4806** | 40 | 92% | 8250 |
| | Toeplitz | 2525 | 103 | 85% | 4608 |
| | Circulant | 1728 | 82 | 88% | 3936 |
| | Unstructured | 3614 | 943 | 0% | 31488 |
| Hopper | Chromatic | **3408** | 11 | 92% | 3960 |
| | Masked | 2196 | 17 | 91% | 4726 |
| | Toeplitz | **2749** | 94 | 78% | 4320 |
| | Circulant | 2680 | 82 | 80% | 3936 |
| | Unstructured | 2691 | 574 | 0% | 19680 |
| Walker2d | Chromatic | **3695** | 17 | 94% | 6571 |
| | Masked | 1781 | 19 | 94% | 6635 |
| | Toeplitz | 1 | 103 | 85% | 4608 |
| | Circulant | 3 | 82 | 88% | 3936 |
| | Unstructured | **2230** | 943 | 0% | 31488 |

Table 2: Comparison of the best policies from five distinct classes of RL networks: chromatic (ours), masked (networks from Subsection 4.1), Toeplitz networks from (Choromanski et al., 2018), circulant networks and unstructured trained with standard ES algorithm (Salimans et al., 2017). All results are for feedforward nets with one hidden layer of size $h = 41$.

Our experiments show that these approaches fail by producing suboptimal policies for harder tasks (see: Fig. 4). Note also that if a partitioning distribution is fixed throughout the entire optimization, training policies for such tasks and restarting to fix another partitioning or distribution can cause substantial waste of computational resources, especially for the environments requiring long training time. However we also notice an intriguing fact that random partitionings still lead to nontrivial rewards. We find it similar to the conclusions in NAS for supervised learning (Pham et al., 2018) - while training a random child model sampled from a reasonable search space may produce $\geq$

80 % accuracy, the most gains from a controller will ultimately be at the tail end; i.e. at the 95% accuracies.

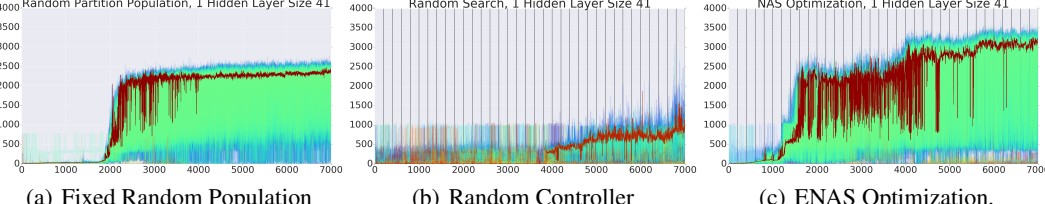

(a) Fixed Random Population      (b) Random Controller      (c) ENAS Optimization.

Figure 4: Random partitioning experiments versus ENAS for Walker2d. Curves of different colors correspond to different workers. The maximal obtained rewards for random partitionings/distributions are smaller than for chromatic networks by about 1000. (a): Fixed random population of 301 partitioning for joint training. (b): Replacing the ENAS population sampler with random agent. (c): Training with ENAS.

We observe that by training with ENAS, the entire optimization benefits in two ways by: **(1)** selecting partitionings leading to good rewards, **(2)** resampling good partitionings based on the controller replay buffer, and breaking through local minima inherently caused by the weight-sharing mechanism maximizing average reward. We see these benefits precisely when a new ENAS iteration abruptly increases the reward by a large amount, which we present on Fig. 5.

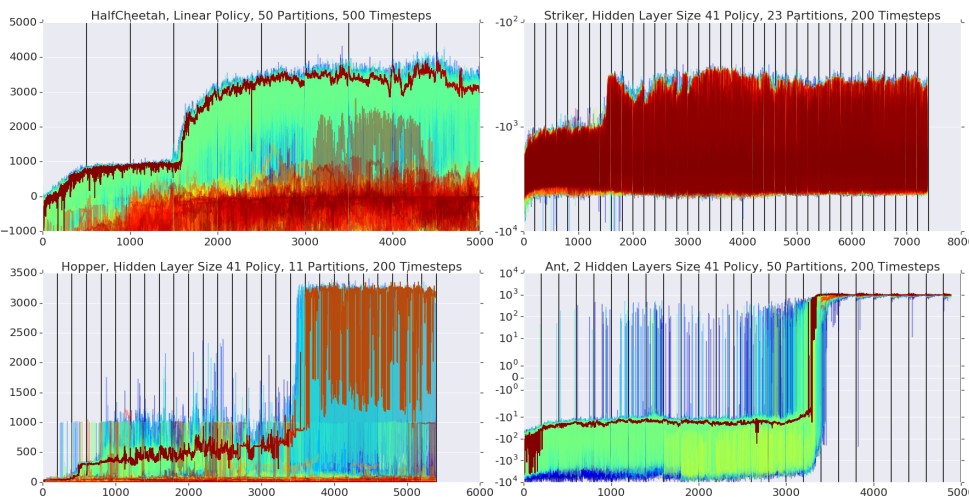

Figure 5: Training curves for four OpenAI Gym environments: HalfCheetah, Striker, Hopper and Ant. Timesteps when training curves abruptly increase are correlated with those when ENAS controller produces new partitioning suggestions (that are marked by black vertical lines).

## 5   CONCLUSION & FUTURE WORK

We presented a principled and flexible algorithm for learning structured neural network architectures for RL policies and encoded by compact sets of parameters. Our architectures, called chromatic networks, rely on partitionings of a small sets of weights learned via ENAS methods. Furthermore, we have also provided a scalable way of performing NAS techniques with RL policies which is not limited to weight-sharing, but can potentially also be used to construct several other combinatorial structures in a flexible fashion, such as node deletions and edge removals.

We showed that chromatic networks provide more aggressive compression than their state-of-the-art counterparts while preserving efficiency of the learned policies. We believe that our work opens new research directions, especially from using other combinatorial objects. Detailed analysis of obtained partitionings (see: Appendix C) also shows that learned structured matrices are very different from previously used state-of-the-art (in particular they are characterized by high displacement rank), yet it is not known what their properties are. It would be also important to understand how transferable those learned partitionings are across different RL tasks (see: Appendix D).

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

APPENDIX

## A  FULL PLOTS

### A.1  TRAINING CURVES

We plot black vertical bars in order to denote a NAS update iteration. Different curves correspond to different workers.

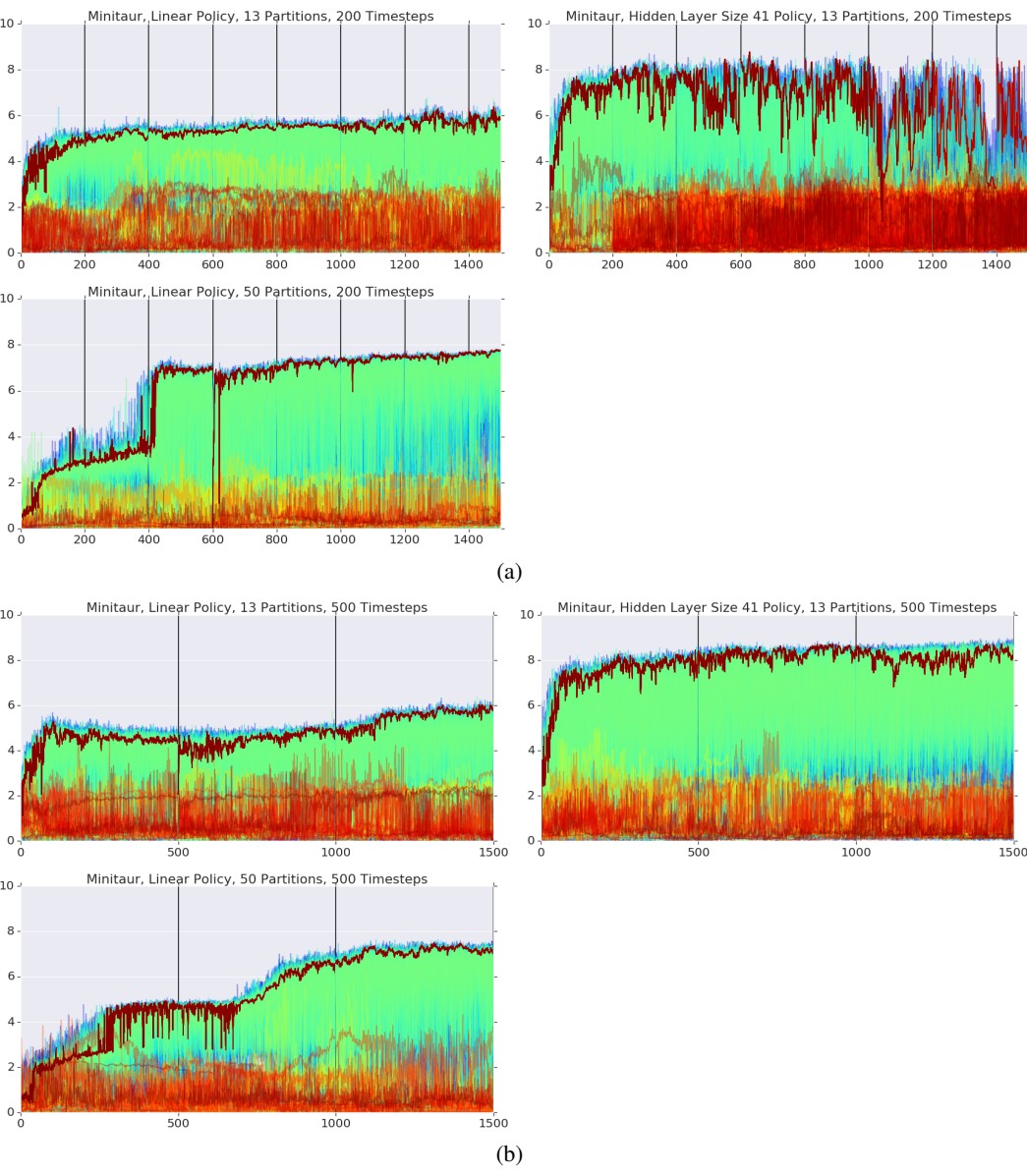

Figure 6: Minitaur Ablation Studies

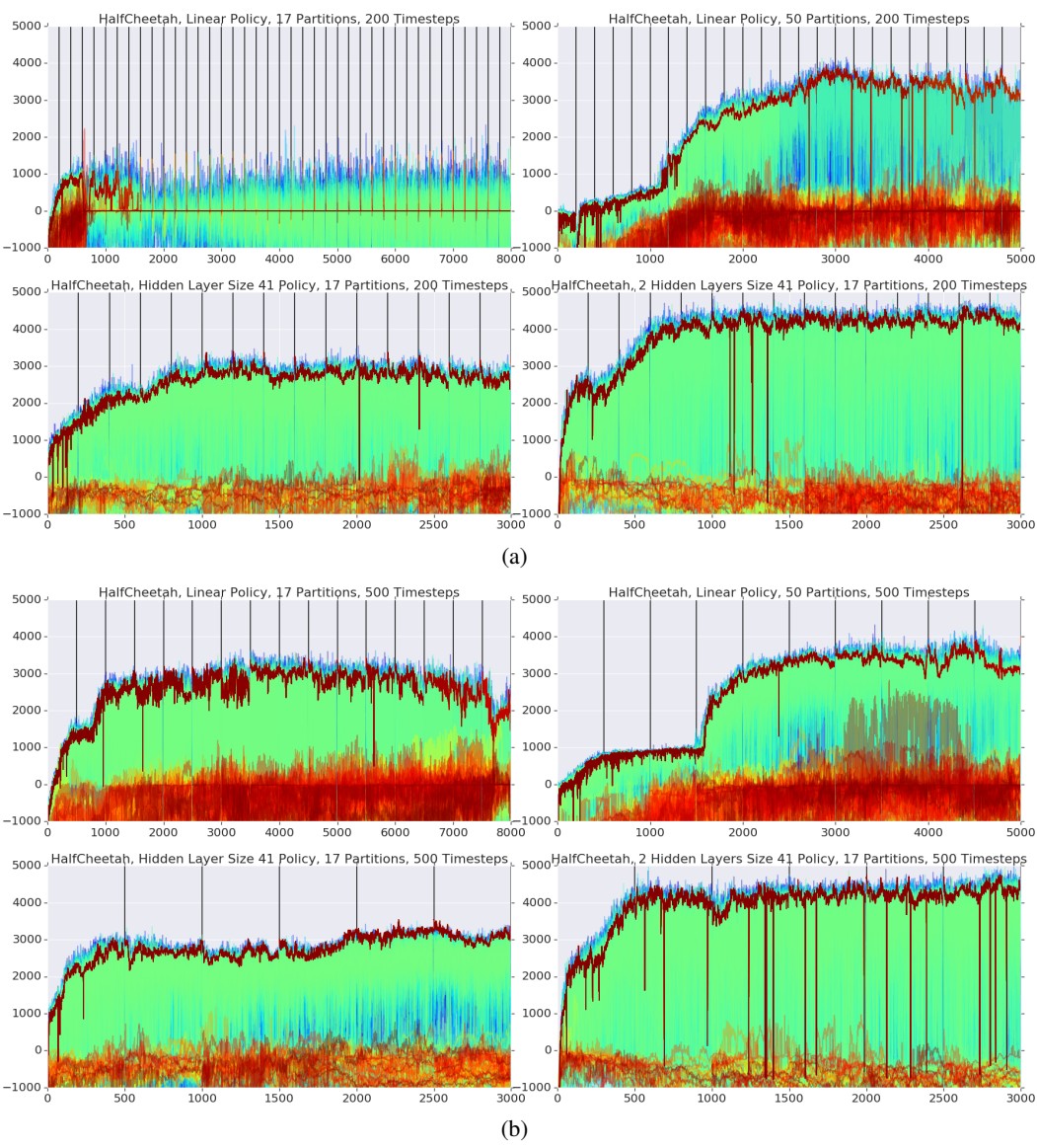

Figure 7: HalfCheetah Ablation Studies

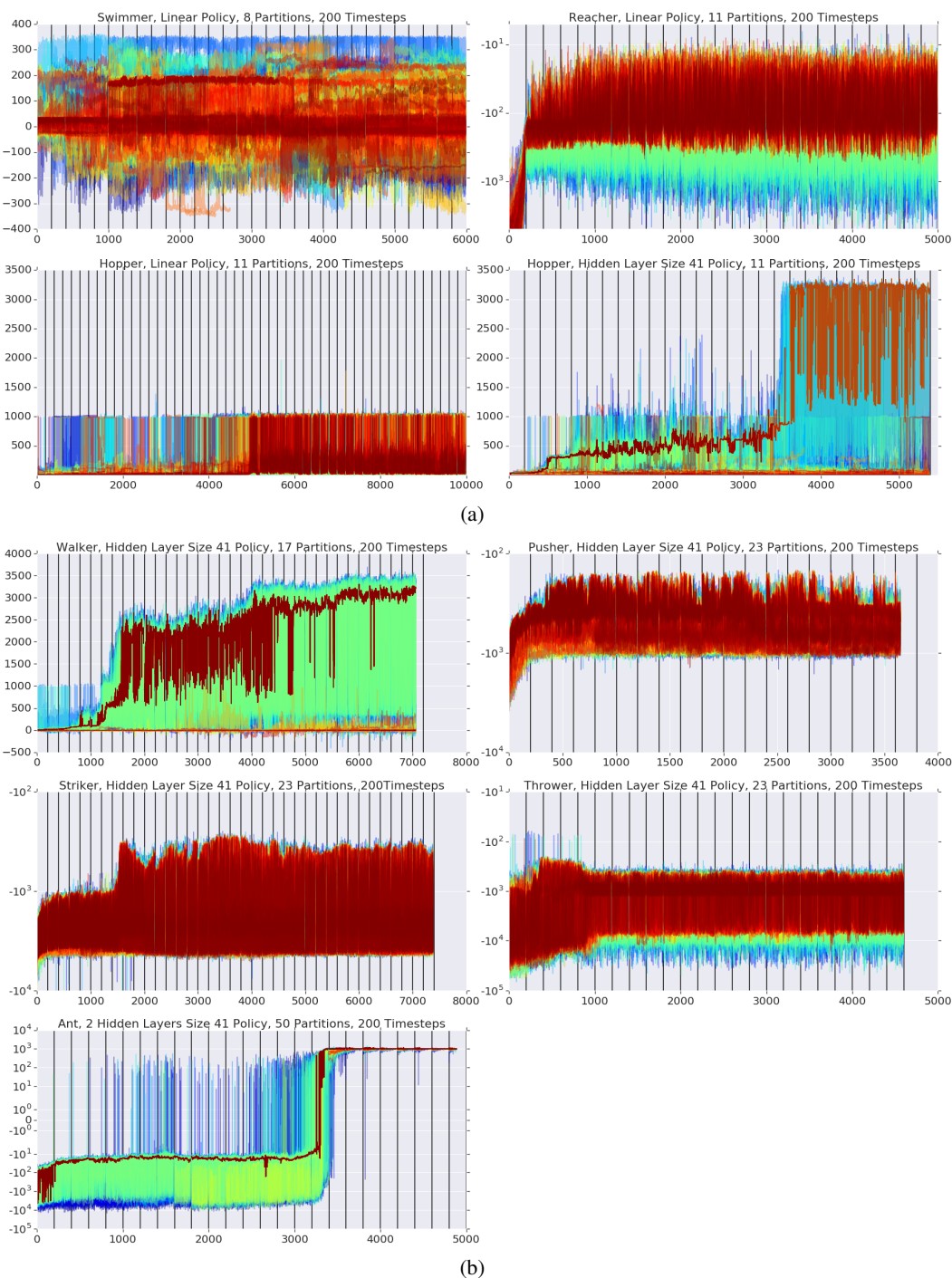

Figure 8: Other environments (Swimmer, Reacher, Hopper, Walker, Pusher, Striker, Thrower, Ant)

## A.2 Maximum Reward Curves

In order to view the maximum rewards achieved during the training process, for each worker at every NAS iteration, we record the maximum reward within the interval [NAS_iteration $\cdot T$, (NAS_iteration $+1) \cdot T$), where $T$ stadns for the current number of conducted timestpes.

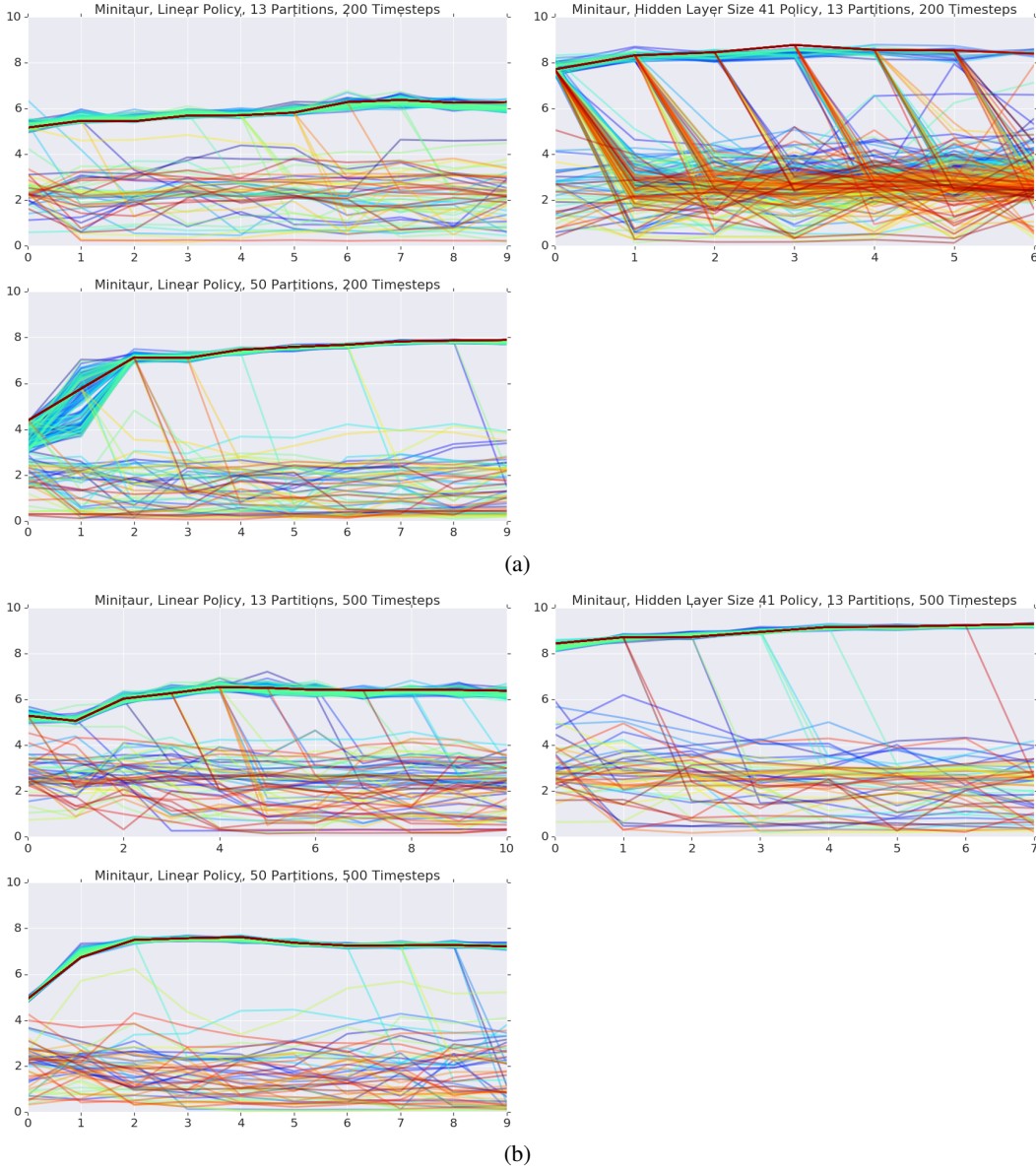

Figure 9: Maximum Reward Curves - Minitaur Ablation Studies

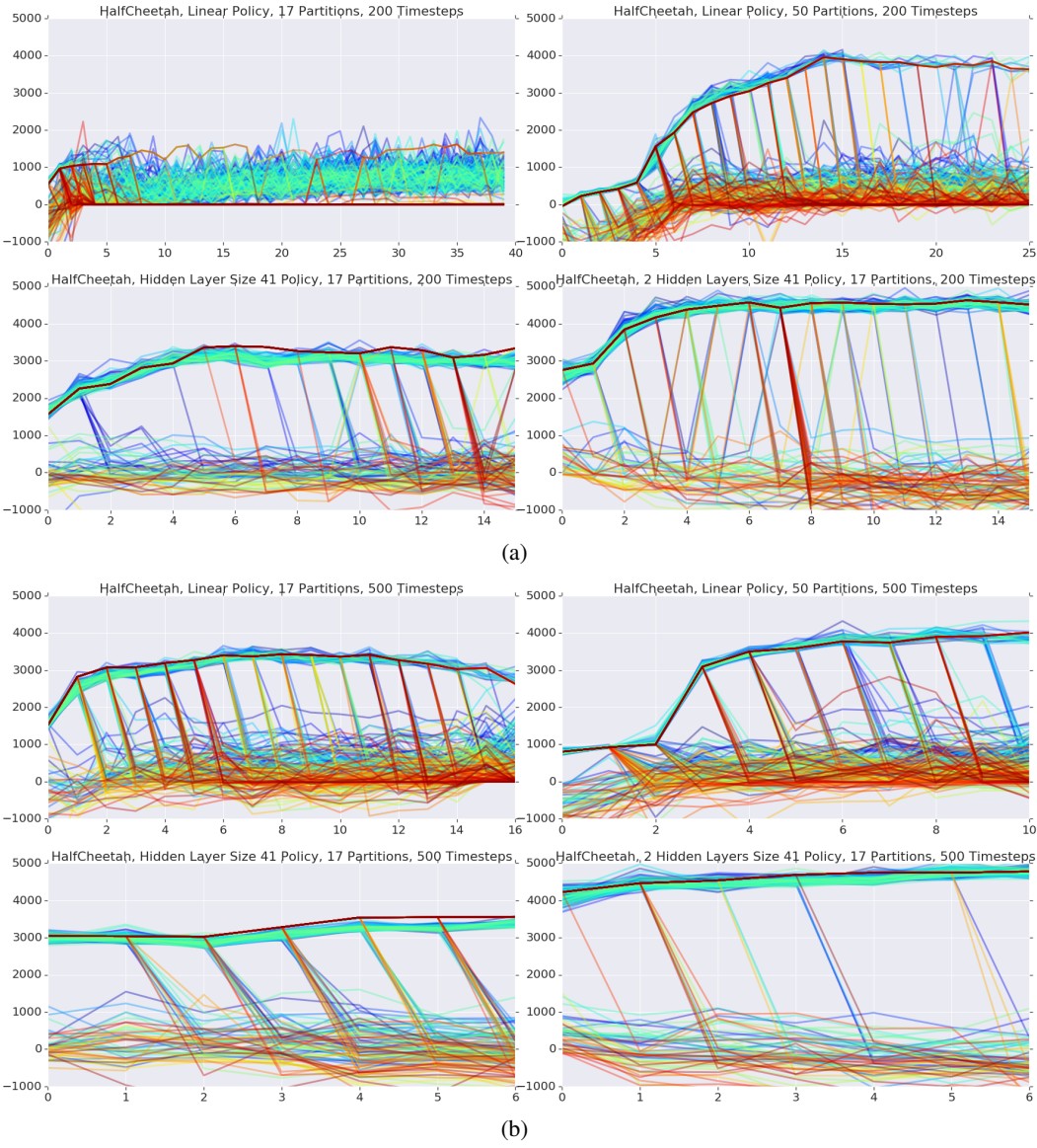

Figure 10: Maximum Reward Curves - HalfCheetah Ablation Studies

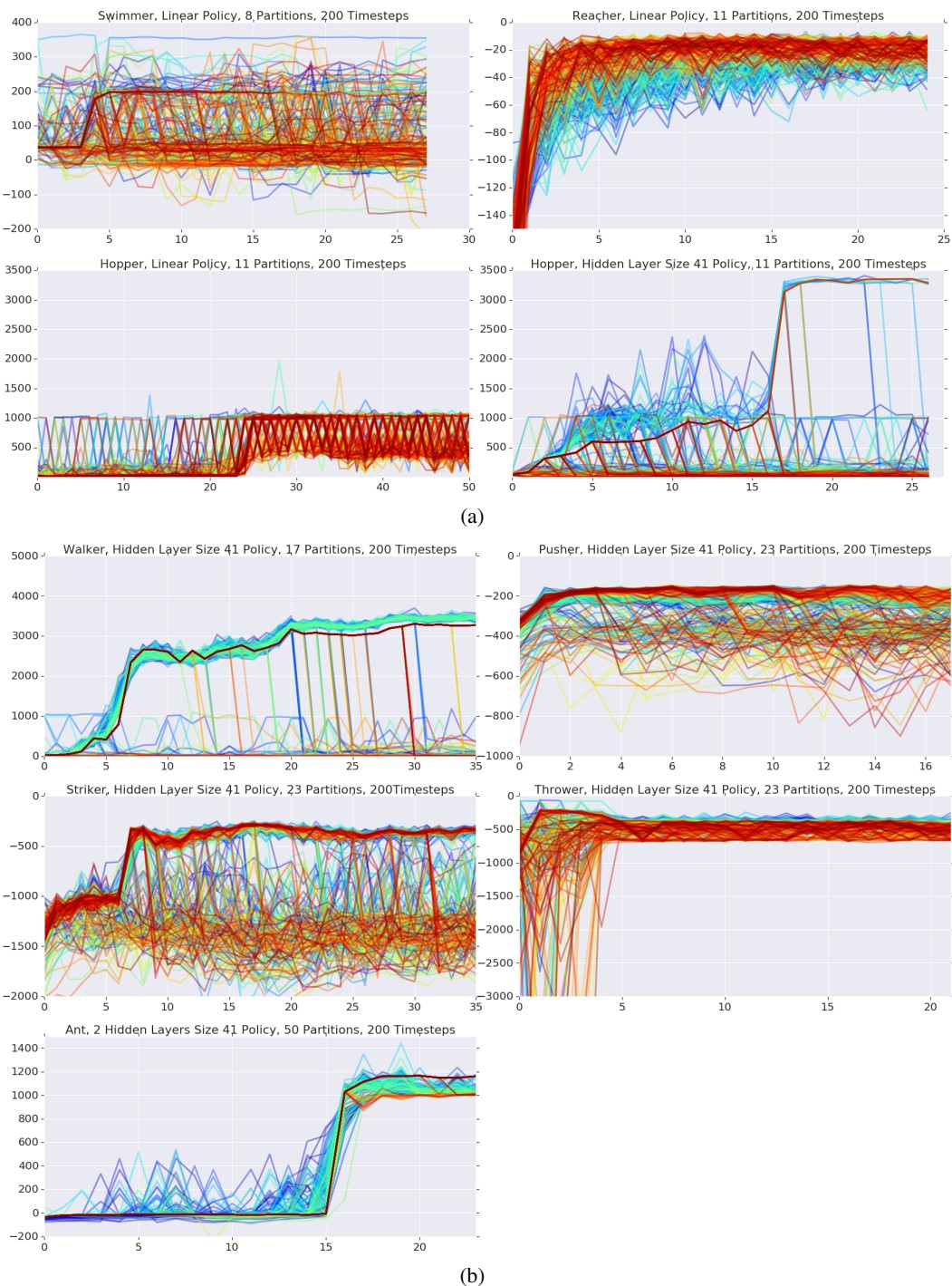

Figure 11: Maximum Reward Curves - other environments (Swimmer, Reacher, Hopper, Walker, Pusher, Striker, Thrower, Ant)

## B   EXACT SETUP AND HYPERPARAMETERS

### B.1   CONTROLLER SETUP

We set LSTM hidden layer size to be 64, with 1 hidden layer. The learning rate was $0.001$, and the entropy penalty strength was $0.3$. We used a moving average weight of $0.99$ for the critic, and used a temperature of $1.0$ for softmax, with the training algorithm as REINFORCE.

### B.2   POLICY

We use tanh non-linearities. For all the environments, we used reward normalization, and state normalization from (Mania et al., 2018a) except for Swimmer. We further used action normalization for the Minitaur tasks.

### B.3   ES-ALGORITHM

For ES-optimization, we used algorithm from (Choromanski et al., 2018), where we applied Monte Carlo estimators of the Gaussian smoothings to conduct gradient steps using forward finite-difference expressions. Smoothing parameter $\sigma$ and learning rate $\eta$ were: $\sigma = 0.1, \eta = 0.01$.

## C   PARTITIONINGS METRICS

We analyzed obtained partitionings to understand whether they admit simpler representations and how they relate to the partitionings corresponding to Toeplitz-like matrices that are used on a regular basis for the compactification of RL policies.

### C.1   SINGLE PARTITIONING METRICS

**Partitionings' Entropies:**   We analyze the distribution of color assignments for network edges for a partitioning, by interpreting the number of edges assigned for each color as a count, and therefore a probability after normalizing by the total number of edges. We computed entropies of the corresponding probabilistic distributions encoding frequencies of particular colors. We noticed that entropies are large, in particular the representations will not substantially benefit from further compacification using Huffman coding (see: Fig. 12).

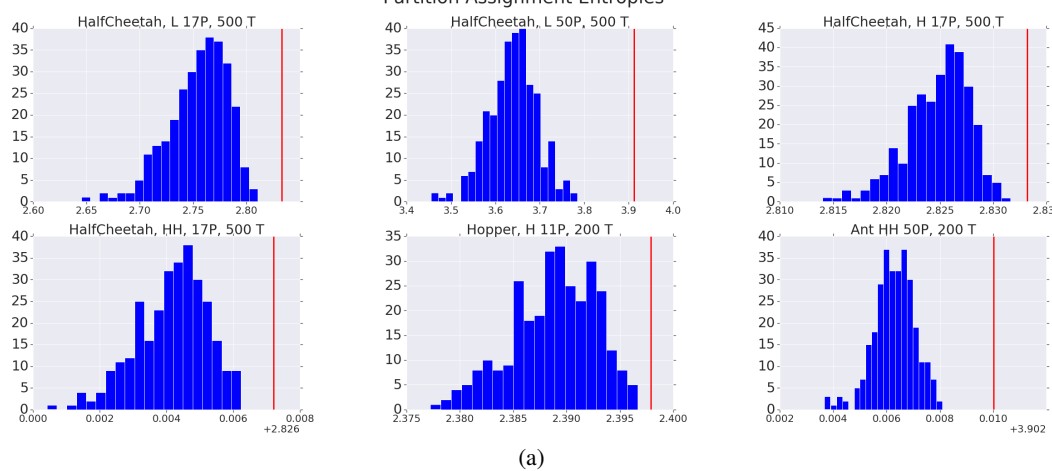

(a)

Figure 12: The blue bars count the number of produced partitionings with the entropy within given range. For the comparison the entropy of the random uniform partitioning is presented as a red line.

**Displacement Rank:** Recall that the displacement rank (Sindhwani et al., 2015; Demmel & Koev) of a matrix $R$ with respect to two matrices: $F, A$ is defined as the rank of the resulting matrix $\nabla_{F,A}(R) = FR - RA$. Toeplitz-type matrices are defined as those that have displacement rank 2 with respect to specific band matrices (Sindhwani et al., 2015; Demmel & Koev). We further analyze the displacement ranks of the weight matrices for our chromatic networks, and find that they are full displacement rank using band matrices $(F, A)$ for both the Toeplitz- and the Toeplitz-Hankel-type matrices (see: Fig. 13).

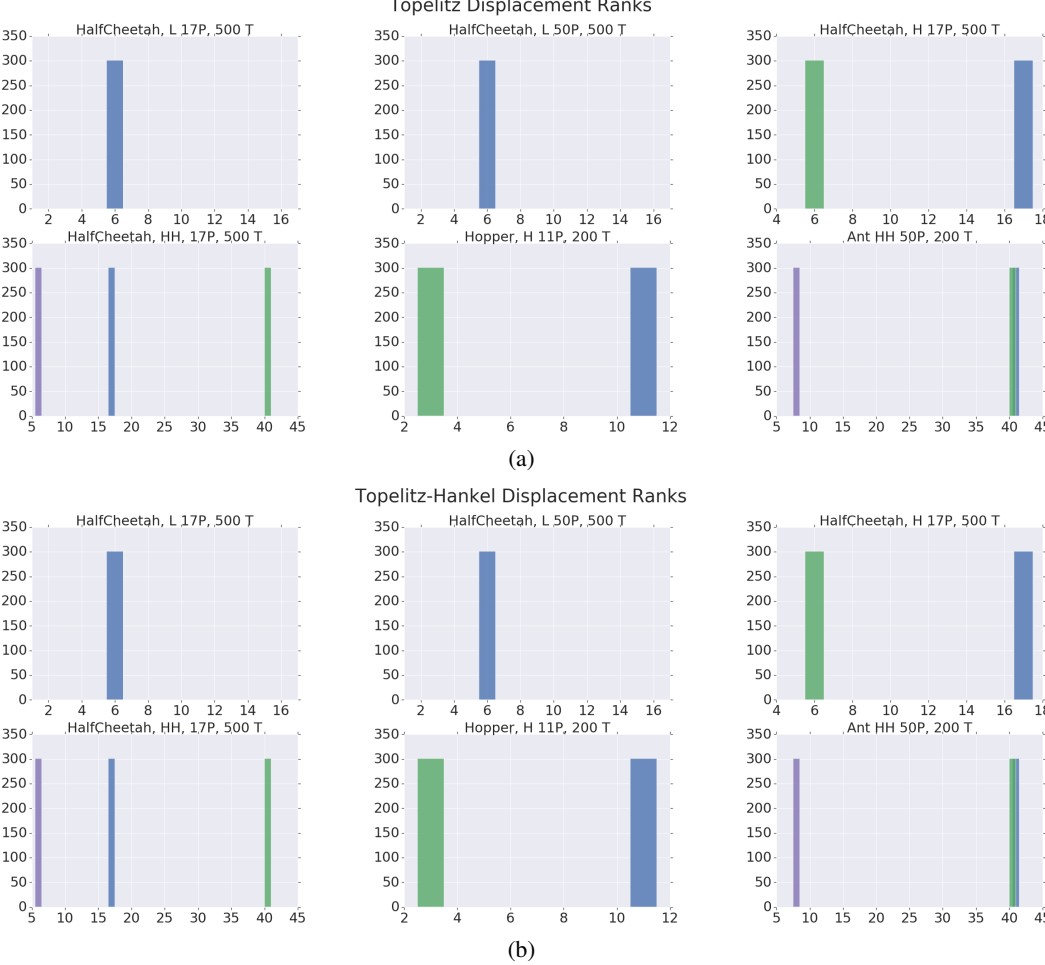

Figure 13: Displacement Ranks of Weight Matrices induced by Partitions at the end of training. We round an entry of the matrix to $0$ if its absolute value is less than $0.1$.

### C.2 Pairwise Partitionings Metrics

We examine the partitionings produced by the controller throughout the optimization by defining different metrics in the space of the partitionings and analyzing convergence of the sequences of produced partitionings in these matrics. We view partitionings as clusterings in the space of all edges of the network. Thus we can use standard cluster similarity metrics such as RandIndex (Rand, 1971) and Variation of Information (Meilă, 2003). Distance metric counts the number of edges that reside in different clusters (indexed with the indices of the vector of distinct weights) in two compared partitionings/clusterings. We do not observe any convergence in the analyzed metrics (see: Fig.14). This suggests that the space of the partitionings found in training is more complex. We leave its analysis for future work.

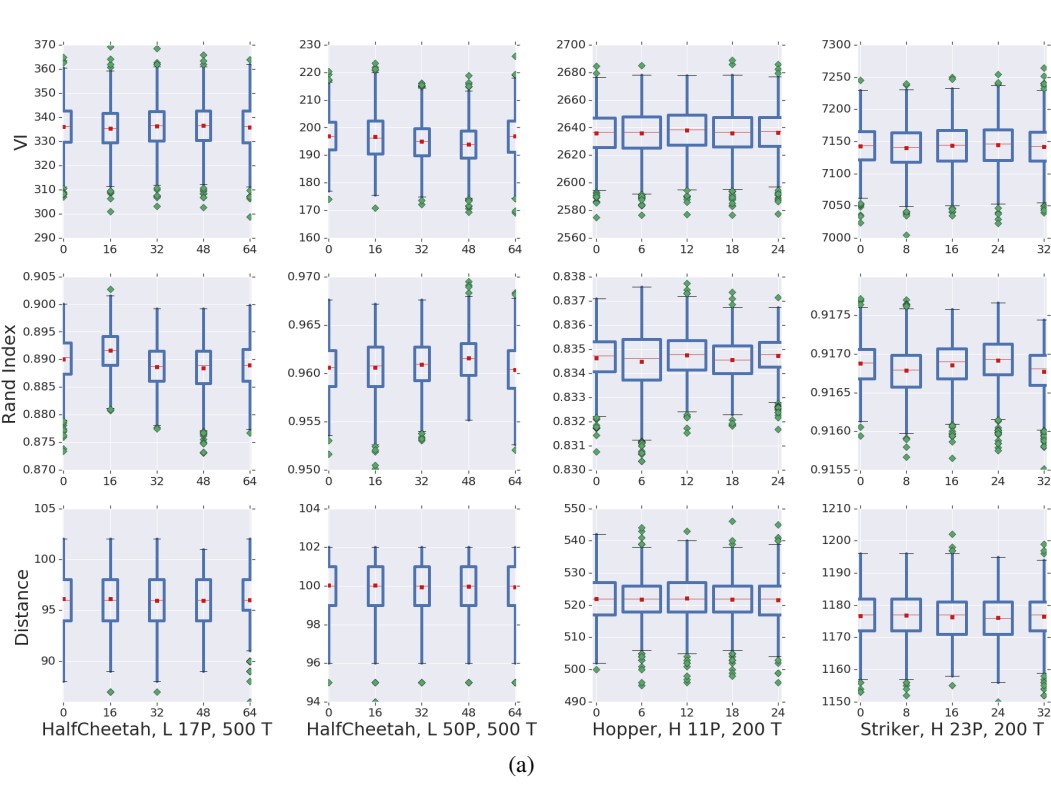

Figure 14: Variation of Information (VI), RandIndex, and Distance. X-axis corresponds to NAS iteration number during training.

## D    TRANSFERABILITY

We tested transferability of partitionings across different RL tasks by using the top-5 partitionings (based on maximal reward) from HalfCheetah (one-hidden-layer network of size $h = 41$), and using them to train distinct weights (for the inherited partitionings) using vanilla-ES for Walker2d (both environments have state and action vectors of the same sizes). This is compared with the results when random partitionings were applied. Results are presented on Fig.15. We notice that transfered partitionings do not underperform. We leave understanding the scale in which these learned partitionings can be transfered across tasks to future work.

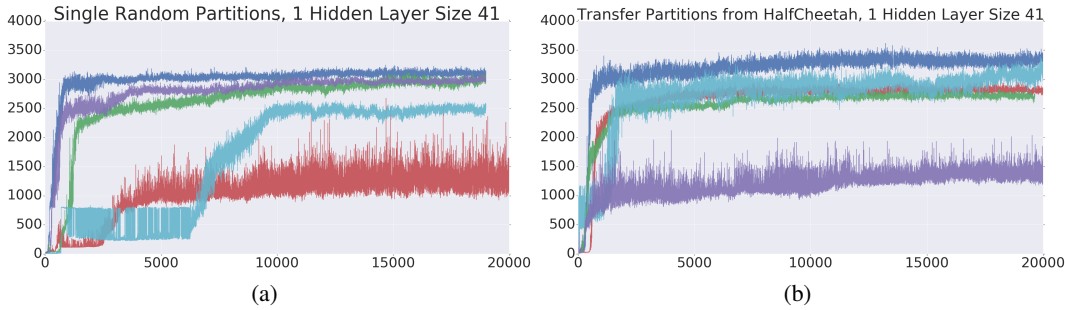

Figure 15: (a): Random partitioning used to train Walker2d. (b): Transfer of the partitioning from HalfCheetah to Walker2d. Transfered partitionings do not underperform.

## E    EXPERIMENT SETUP FOR BASELINES IN TABLE 2

We compare the Chromatic network with other established frameworks for structrued neural network architectures. In particular, we consider Unstructured, Toeplitz, Circulant and a masking mechanism (Choromanski et al., 2018; Lenc et al., 2019). We introduce their details below.

Notice that all baseline networks share the same general architecture: 1-hidden layer with $h = 41$ units and tanh non-linear activation. To be concrete, we only have two weight matrices $W_1 \in \mathbb{R}^{|\mathcal{S}| \times h}, W_2 \in \mathbb{R}^{h \times |\mathcal{A}|}$ and two bias vectors $b_1 \in \mathbb{R}^h, b_2 \in \mathbb{R}^{|\mathcal{A}|}$, where $|\mathcal{S}|, |\mathcal{A}|$ are dimensions of state/action spaces. These networks differ in how they parameterize the weight matrices.

**Unstructured.**    A fully-connected layer with unstructured weight matrix $W \in \mathbb{R}^{a \times b}$ has a total of $ab$ independent parameters.

**Toeplitz.**    A toeplitz weight matrix $W \in \mathbb{R}^{a \times b}$ has a total of $a+b-1$ independent parameters. This architecture has been shown to be effective in generating good performance on benchmark tasks yet compressing parameters (Choromanski et al., 2018).

**Circulant.**    A circulant weight matrix $W \in \mathbb{R}^{a \times b}$ is defined for square matrices $a = b$. We generalize this definition by considering a square matrix of size $n \times n$ where $n = \max\{a, b\}$ and then do a proper truncation. This produces $n$ independent parameters.

**Masking.**    One additional technique for reducing the number of independent parameters in a weight matrix is to mask out redundant parameters (Lenc et al., 2019). This slightly differs from the other aforementioned architectures since these other architectures allow for parameter sharing while the masking mechanism carries out pruning. To be concrete, we consider a fully-connected matrix $W \in \mathbb{R}^{a \times b}$ with $ab$ independent parameters. We also setup another mask weight $S \in \mathbb{R}^{a \times b}$. Then the mask is generated via

$$M = \text{softmax}(M/\alpha),$$

where softmax is applied elementwise and $\alpha$ is a constant. We set $\alpha = 0.01$ so that the softmax is effectively a thresolding function wich outputs near binary masks. We then treat the entire concatenated parameter $\theta = [W, S]$ as trainable parameters and optimize both using ES methods. At

convergence, the effective number of parameter is $ab \cdot \eta$ where $\eta$ is the proportion of $M$ components that are non-zero. During optimization, we implement a simple heuristics that encourage sparse network: while maximize the true environment return $R = \sum_t r_t$, we also maximize the number of mask entries that are zero $1 - \eta$. The ultimate ES objective is: $R' = \beta \cdot R + (1 - \beta) \cdot (1 - \eta)$, where $\beta \in [0, 1]$ is a combination coefficient which we anneal as training progresses. We also properly normalize $R$ and $(1 - \eta)$ before the linear combination to ensure that the procedure is not sensitive to reward scaling.

