# OpenReview forum: "Reinforcement Learning with Chromatic Networks"
_ICLR.cc/2020/Conference — Reject_

### Official Review · AnonReviewer2 · 2019-10-22
**Official Blind Review #2**

**Rating:** 3

**Review:**

The authors propose to construct reinforcement learning policies with very few parameters. For this purpose, they force a feed-forward neural network to share most of its weights, reducing the total number of different weights to at most 23 and therefore compress the network. Instead of manually encoding which weights are shared, the authors propose to use a reinforcement learning method to learn this mapping. The values of all parameters are learned with a gradient-based method.

The paper is very well-written. The concepts are easy to follow, the related work covers a lot of different but related domains. The experimental section sheds light on some important aspects. However, I have few concerns regarding Table 2.

Table 1 considers more tasks than Table 2. Why did you decide to use only a subset and why these tasks? The architecture of the FFNN is not part of your search space but you admit it is very important. For the experiments you chose a FFNN with one hidden layer. Furthermore, you manually adapted the number of partitions. Would fixing the architecture to one hidden layer work in general? How do you select the number of partitions? Both these choices seem crucial and to invalidate the automation aspect. You select the best run of 300. Do your baselines follow a comparable setup? What if you compare mean rewards? The discussion of random partitions is very important and it is nice to see that you discuss this in Section 4.3. As you mentioned, in NAS random search is a strong competitor. Therefore, this method deserves to be added to Table 2 as well.
Concluding, this is an okay paper with limited innovation. The comparison to baseline need to be improved or at least justified.

**Experience Assessment:**

I have published one or two papers in this area.

**Review Assessment: Checking Correctness Of Derivations And Theory:**

I assessed the sensibility of the derivations and theory.

**Review Assessment: Checking Correctness Of Experiments:**

I carefully checked the experiments.

**Review Assessment: Thoroughness In Paper Reading:**

I read the paper thoroughly.

---

> ### Author Response · Authors · 2019-11-10
> **Main comments:**
>
> >>  Subset of tasks
> Table 2 is meant to provide a comparison between different variants of network compression, and we believe that the 4 environments provide enough of a representative (in terms of difficulty and diversity) for understanding how our ENAS method performs against the other methods in terms of rewards and compression.
>
> >> Using one hidden layer vs 2 hidden layers:
> We chose one hidden layer as it would provide a strong performance while still lowering inference time. There is a balance here (especially for robotics such as the Minitaur where in many cases, the robot has no on-board GPU and is performing everything on CPU), where linear policies are very fast for inference, but may produce lackluster performance, and 2 hidden layer policies would provide high performance, but are much slower in inference and would slow down the NAS search process.
>
> >> Selecting number of partitions:
> Please see our meta-response.
>
>
> >> Limited Innovation
> Designing compact policies for RL is of crucial importance to research on ES methods for RL and robotics, where such compact policies can be effectively deployed on real hardware. To the best of our knowledge, we are the first to show that such compact structures can be learned (rather than hardcoded as in all previous works on the subject) even though that learning problem is of strongly combinatorial flavor which often makes the task of designing effective and at the same time *scalable* algorithms infeasible.

---

### Official Review · AnonReviewer1 · 2019-10-22
**Official Blind Review #1**

**Rating:** 3

**Review:**

Summary: This paper focuses on neural architecture search for constructing compact RL policies. It combines ideas from the popular ENAS and ES methods for optimisation. Recent work defined a family of compact policies by imposing a fixed Toeplitz structure. This paper introduces the so-called “chromatic network” architecture, which partitions weights of the RL network into tied sub-groups. The partitioning is searched with ENAS and shared weights are updated via ES. Experiments on continuous control benchmarks show that good performance can be obtained using a very small number of parameters. Favourable reward-compression outcomes can be achieved compared to some baseline alternatives.

To my understanding the main contributions are (1) the Chromatic parameter sharing scheme and (2) the combined ENAS+ES learning procedure.

Pros:
+ Chromatic networks provide a neat idea for managing compact networks via parameter sharing. (Although not dramatically novel given cited work by Salimans, Gaier, etc)
+ Searching both the partitioning + weight values with ENAS and ES provides a nice way to do learning & compact architecture search simultaneously.
+ Results generally show favourable compression/reward performance vs Mask, Toeplitz  and Circulant baselines.

Cons & Questions:
0. Motivation. The paper claims that RL can be high-dimensional with millions of parameters, and so there is a need to apply NAS to RL. But experiments are conducted in low dimensional environments with only 100s of parameters. The empirical validation doesn’t match the motivating scenario. We would want to see results on successful compression of larger vision-based networks to be fully persuaded. Without this it undermines the significance of the paper. The paper makes a claim about embedded devices, but this is unconvincing, as for these proprioceptive control tasks, the uncompressed networks are already small enough to run on most embedded devices.

 1. ENAS vs Chromatic. Both use RL controllers. The main difference to ENAS seems to be the Chromatic sharing scheme, and use of ES rather than Backprop to update the weights. Some points of motivation/justification are not very clear after reading: (1) Why vanilla ENAS can’t be used for RL? (with the modification of replacing standard backdrop with any standard continuous control RL algorithm for weight updates). It is sort of claimed that ENAS can’t be used, it’s not obvious to me that this is true. (2) How is the  Chromatic strategy specific for RL? In principle it should also be usable in SL. Assessing its performance in SL would be more broadly relevant, interesting and significant since there is more prior compression work in SL.

2. The paper is a bit vague about how number of partitions is set. Is it a user-specifiable parameter, if so how do we set it? It’s suggested that partition number is included in the reward function, but then it’s even more unclear how to calibrate this to hit a particular performance or size target (or optimize performance for fixed size, size for fixed performance constraint) that a user may require in practice.

3. The real (wall-clock) running time for different NAS methods should be reported in the results.

4. Baselines. The presented results are a reasonable start, but they are all very much variations within the same family. One would like to know how the current method compares to: (i) Direct application of ENAS (See also Q1), (ii) Training the full network and applying standard NN pruning techniques (such as magnitude based), (iii) Baseline of training a comparably small sized network directly. (ivv) Importantly, since the size of networks here are very small, classic neuro-evolution algorithms from evolutionary robotics such as NEAT (Stanley & Miikkulainen) seem to provide a reasonable alternative. It’s hard to know if to be impressed with the current result or not without seeing the results for a decent NEAT-like competitor. (See also Q0).

Other:
A. A presentation of the algorithm with pseudocode would help the reader follow an overview of the algorithm.
B. Is the RL network/method used for the tasks the same as the network used in paper “Structured Evolution with Compact Architectures for Scalable Policy Optimization”? If so what is the source of the discrepancy between Tab 2 here and Tab 1 in that paper?
C. The title doesn’t make the content obvious to a naive prospective reader. At least something like “RL with Chromatic Networks for Compact Architecture Search” would be more informative.
D. Eq (1) has a vspace error.


**Experience Assessment:**

I have published one or two papers in this area.

**Review Assessment: Checking Correctness Of Derivations And Theory:**

I assessed the sensibility of the derivations and theory.

**Review Assessment: Checking Correctness Of Experiments:**

I assessed the sensibility of the experiments.

**Review Assessment: Thoroughness In Paper Reading:**

I read the paper at least twice and used my best judgement in assessing the paper.

---

> ### Author Response · Authors · 2019-11-10
> **Main comments - part I:**
>
> Thank you for your comments.
>
> >> Motivation
> The main topic of this paper are compact neural networks encoding RL policies which is of key interest to roboticians deploying their policies on real hardware with limited storage.  For instance, the Minitaur [1] does not have an onboard GPU yet must walk in real life with very little delay between actions, and thus lowering the load on a basic CPU is crucial. Thus we think that the topic is actually much more relevant to robotics than to general RL or ML. While the literature on the compactification of neural nets is voluminous, as we explain in the introduction, most of the existing methods cannot be adapted to the RL setting. For instance, quantization methods compactifying the network after training and used on a regular basis for compressing ML models, are too crude for a subtle task of finding compact representations for RL policies.
>
> We also do not agree that we do not support our claims with tested scenarios. Standard ES algorithms for learning OpenAI gym tasks require few thousand parameters (even for tasks such as Swimmer, see Salimans et al., where two hidden layers of size 32-64 are used). Recent papers suggest that that number of parameters can be reduced to few hundred by low displacement rank policies. We show that we can obtain even more aggressive reduction and learn with fewer than 20 parameters. That does not only mean reducing sampling complexity of ES methods. For tasks such as locomotion for minitaur, where our policies can be deployed on real hardware (and are comparable state-of-the-art  ones trained by ARS algorithm) such a compactification of parameters means much longer runs (typical ones for 100-parameter policies for Ghost Robotics minitaurs do not last more than 10-15 minutes between consecutive battery recharging phases).
>
> We want to raise several points here about the case for robotics. Firstly, as we have mentioned above, small policy functions are of key interest to roboticians deploying their policies on real hardware with very limited storage constraints. Secondly, in plenty of robotics applications off-robot training time is much less critical than the achieved compactification levels. For instance, roboticians often perform highly-distributed on-policy training using simulators and then boost the performance of their policies (effectively reducing sim-to-real gap) by conducting more rollouts on the real hardware. Learning compact efficient policies that could be good initial points for on-robot training is then critical. Policies parameterized by few params provide a way to substantially simplify on-hardware training by reducing the number of *real* on-robot rollouts than need to be conducted and that in practice are the most time-consuming part. Finally, as explained in “Inference Time” in Section 4.2, chromatic networks provide *subquadratic* inference time as opposed to quadratic as for standard networks which results in more efficient energy-usage.
>
> All these raised points are of great importance to robotics, which we think is one of the primary benefits of this work.
>
> >> Why ENAS can’t be used for general RL/other RL algorithms
> To address the concern about using other methods such as Q-learning or Policy Gradient, we first note that to date, there has not been previous work on combining these specific methods with NAS search. In particular for ENAS and chromatic networks, it is incredibly difficult to setup a connection between the object that the ENAS controller outputs (e.g. a partition) to the policy parameters, in a completely differentiable end-to-end way. In general, the combinatorial objects that the ENAS controller produces are infeasible to be sent through in terms of a computation graph. Instead, ENAS assumes that the outputs it generates will have a (non-differentiable, numpy) reward associated with them and hence why the ENAS controller requires a policy gradient update between its output and the reward.
>
> With this setting in mind, the ENAS controller would still require a numpy reward associated to its output - while it is possible to perform ENAS with rewards sent by Q-learning or policy gradient, the reward signals would be far fewer compared to ES as normally Q-learning/policy gradient setups are based on a single centralized set of policy parameters, whereas ES uses an extensive wide range of policy parameters, an assumption ENAS is built upon.

---

> > ### Author Response · Authors · 2019-11-10
> > **Main comments - part II:**
> >
> > >> Chromatic Strategy specific for RL vs SL
> > We believe our method is most natural and specialized for RL tasks, as it combines ES and ENAS in a highly scalable way. To give context, vanilla NAS [Zoph et al, 2018] for classical supervised learning setting (SL) requires a large population of ~450 GPU-workers (“child models”) all training one-by-one, which results in many GPU-hours of training. ENAS uses weight sharing across multiple workers to reduce the time, although it can reduce computational resources at the cost of the variance of the controller’s gradient. Our method solves both issues (fast training time and low controller gradient variance) by leveraging a large population of much-cheaper CPU workers (300) increasing the effective batch-size of the controller, while also training the workers simultaneously via ES. This setup is not possible in SL, as single CPUs cannot train large image-based classifiers in practice.
> >
> > >> Number of partitions to set
> > Please see our meta-response.
> >
> > >> Wall clock time
> > We did not find a significant difference in wall clock time of training, as all of the methods essentially the same speed as vanilla ES. In particular, because the policy only uses forward passes and uses only NumPy operations, the forward propagation from state -> action using the same architecture takes the same time on a CPU worker. The policies for Circulant/Toeplitz/Masking all use Python dictionaries to map between shared weights -> actual weights, which does not incur extra overhead compared to using the actual weights for an unstructured policy from matrix memory. Assuming the ENAS controller runs on a fast GPU (thus there is very little delay for the master worker which propagates the partitions), ENAS also follows the same principle.
> >
> > >> Baselines
> > As discussed earlier, (i) Direct ENAS would not be feasible given the controller cannot be tractibly connected to the policy end-to-end. (ii) We believe that our masking procedure is very similar to this approach. (iii) We believe that reducing the hidden layer size (while maintaining good performance) while still producing comparable bit-wise compression is already not possible. This is because if the action dimension is D, then having a hidden layer of the same size D would still incur at least more parameters than a linear policy. Since on average, the action dimensionality D~5`, a hidden layer of size ~5 is highly likely to perform poorly as well. (ivv) We will take this into consideration in the final version of the paper. But as mentioned in our motivations, it may be difficult to generate an alternate competitive combinatorial search space (than weight sharing) that is simple enough to achieve strong compression while still maintaining high reward.
> >
> > Other:
> > A. Thank you for the comment, we will add a pseudocode in the final version.
> > B. The Toeplitz policies are based on work from that referenced paper so the architecture is the same but other metaparameters are different (number of hidden layers, normalization methods, etc.)
> > C. Thank you for the suggestion - we will strongly consider this title afterwards, as indeed our target is towards practical robotics uses.
> > D. Fixed, thanks.

---

### Official Review · AnonReviewer3 · 2019-10-23
**Official Blind Review #3**

**Rating:** 6

**Review:**

This paper compresses policy networks using approaches inspired by neural architecture search. The main idea is to have a fixed size weight matrix, but learn how to share weights, so that the resulting network can be compressed by storing only unique weights values. Both the partitioning and weights are trained simultaneously, inspired by ENAS. The partitioning is modeled by an autoregressive RNN and trained via REINFORCE. The weights are modeled by a single set of weights (as opposed to a distribution) which is then updated by using a gradient approximation based on ES. The experiments carried out include comparing to existing works on policy network compression, ablating against random partitioning, as well as a few experiments meant to increase understanding of the learned partitions.

Pros:
 - Overall, paper is well-presented and is generally quite clear on its contributions, place in the literature, and experiment details.
 - The approach is straightforward and has applications in compressing RL policies.
Cons:
 - There should be standard deviations in Table 2 across multiple seeds, as it is unclear whether the differences in reward are significant.
- There is some discussion on scalability in the introduction as part of the motivation, though would be nice to see some quantitative numbers. Is the current method already highly scalable, or is scalable still a potential that has yet to be reached?

Questions:
 - What is the computational cost in total CPU time, compared to baselines such as fixed random partitioning? The autoregressive sampling of partitioning seems to be a bottleneck during training, as sampling may be slow, especially when scaling to larger weight matrices. Is this why the partitioning is updated much more sparsely (as shown in Figure 5)?
 - For the gradient estimation in eq 3, what exactly is this unbiased with respect to? Did you mean asymptotically unbiased in the limit as sigma -> 0?
 - Do Chromatic networks always use M partitions? (How exactly were the number of partitions chosen, for Tables 1 and 2?) If I understood correctly, Figure 3 is about the Masked networks from (Lenc et al., 2019). If so, is there a similar method in which the number of partitions of Chromatic networks can be learned (or regularized)?
 - (I may have misunderstood these figures, but:) There seems to be quite a large difference in training curves for different workers, e.g. shown in Figures 4 and 5. The red curve seems to almost always be best, while the green curve is much higher variance. Why is this? I also couldn't tell exactly how many colors there are in these plots (introduction mentions 300?), but wouldn't a mean/std plot be easier to parse (or do the specific colors mean something significant)?
 - Is there a reason why Toeplitz and Circulant have missing numbers in the #bits columns in Table 2?

**Experience Assessment:**

I have read many papers in this area.

**Review Assessment: Checking Correctness Of Derivations And Theory:**

N/A

**Review Assessment: Checking Correctness Of Experiments:**

I assessed the sensibility of the experiments.

**Review Assessment: Thoroughness In Paper Reading:**

I read the paper at least twice and used my best judgement in assessing the paper.

---

> ### Author Response · Authors · 2019-11-10
> **Main comments:**
>
> Thank you very much for all the comments !
>
> >> Scalability of the method: The proposed method is already highly scalable. Proposed algorithm using both ES and ENAS for architecture search is highly parallelizable and can be efficiently distributed across hundreds of machines as we have showed in the paper.
>
> >> CPU time of training: There is almost no delay in comparison to vanilla ES, as the ENAS controller in the central worker can be put into GPU and the output partitions can be parallelized (e.g. batched). The update on the partitions has interval delay as we need to collect the best reward achieved by each worker through the weight sharing mechanism. This best reward becomes more accurate as we increase the interval delay, at the expense of the frequency of partition updates.
>
> >> Table 2: Note that the learned policies are deterministic and by default in OpenAI gym the dynamics is also deterministic. As in many previous papers on the subject (e.g. the one using compact low-displacement rank policies that we compare against) the initial state is also fixed. Thus our results are statistically significant and there is no need to average over many random seeds.
>
> >> Gradient estimation in eq 3
> We meant that normally, the expected value of \g (f(x + \sigma g) - f(x))/\sigma is an unbiased gradient estimate of  E_{g}[f(x + \sigma g)]. We forgot to put in the extra g on the outside; this is corrected now.
>
> >> “Do chromatic networks always use M partitions”
> Please see our meta-response.
>
> >> Toplitz and Circulant have missing bits
> For those two parts, since the corresponding rewards were very low, we decided not to include these results. We have added those numbers.

---

### Author Response · Authors · 2019-11-10
**Meta-response to all reviewers:**

We thank all of the reviewers for their time in reviewing our paper - we appreciate the feedback and have updated our paper. Below we address common questions.

>> Partition Number Setting:
This is a user defined parameter in the algorithm. By default we set the number of partitions to be max(state_dimensionality, action_dimensionality) to enforce that the number of trainable parameters stays linear in comparison to the total number of network weights (which would be e.g. state_dimensionality * action_dimensionality for a linear policy).

---

### Decision · Program_Chairs · 2019-12-19

**Decision:**

Reject

**Comment:**

This paper describes a method for learning compact RL policies suitable for mobile robotic applications with limited storage.  The proposed pipeline is a scalable combination of efficient neural architecture search (ENAS) and evolution strategies (ES).  Empirical evaluations are conducted on various OpenAI Gym and quadruped locomotion tasks, producing policies with as little as 10s of weight parameters, and significantly increased compression-reward trade-offs are obtained relative to some existing compact policies.

Although reviewers appreciated certain aspects of this paper, after the rebuttal period there was no strong support for acceptance and several unsettled points were expressed.  For example, multiple reviewers felt that additional baseline comparisons were warranted to better calibrate performance, e.g., random coloring, wider range of generic compression methods, classic architecture search methods, etc.  Moreover, one reviewer remained concerned that the scope of this work was limited to very tiny model sizes whereby, at least in many cases, running the uncompressed model might be adequate.